# Barriers in care for children with life-threatening conditions: a qualitative interview study in the Netherlands

Marije Brouwer ,[1] Els L M Maeckelberghe,[2] Agnes van der Heide,[3] Irma Hein,[4] Eduard Verhagen[1]

¹Department of Pediatrics, University Medical Center Groningen, University of Groningen, Groningen, The Netherlands
²Institute for Medical Education, University of Groningen, University Medical Center Groningen, Groningen, The Netherlands
³Department of Public Health, Erasmus MC, University Medical Center Rotterdam, Rotterdam, The Netherlands
⁴Psychiatry, Academic Medical Center, Amsterdam, The Netherlands

**Correspondence to**
Marije Brouwer;
m.a.brouwer@umcg.nl

## ABSTRACT

**Objective** To identify barriers, as perceived by parents, to good care for children with life-threatening conditions.
**Design** In a nationwide qualitative study, we held in-depth interviews regarding end-of-life care with parents of children (aged 1 to 12 years) who were living with a life-threatening illness or who had died after a medical trajectory (a maximum of 5 years after the death of the child). Sampling was aimed at obtaining maximum variety for a number of factors. The interviews were transcribed and analysed.
**Setting** The Netherlands.
**Participants** 64 parents of 44 children.
**Results** Parents identified six categories of difficulties that create barriers in the care for children with a life-threatening condition. First, parents wished for more empathetic and open communication about the illness and prognosis. Second, organisational barriers create bureaucratic obstacles and a lack of continuity of care. Third, parents wished for more involvement in decision-making. Fourth, parents wished they had more support from the healthcare team on end-of-life decision-making. Fifth, parents experienced a lack of attention for the family during the illness and after the death of their child. Sixth, parents experienced an overemphasis on symptom-treatment and lack of attention for their child as a person.
**Conclusions** The barriers as perceived by parents focussed almost without exception on non-medical aspects: patient-doctor relationships; communication; decision-making, including end-of-life decision-making; and organisation. The perceived barriers indicate that care for children with a life-threatening condition focusses too much on symptoms and not enough on the human beings behind these symptoms.

## Strengths and limitations of this study

► The variety: Parents from children with various illnesses, ages and prognosis, receiving care from various hospitals were included.
► The size: 64 participating parents from 44 children were interviewed, and over 110 hours of qualitative data on audio were collected.
► The study focusses on young children (aged 1 to 12 years old) and the results cannot be generalised to adolescents or neonates.
► The success in achieving variation in the cultural and religious background of participants with different cultural and religious backgrounds variation was limited.
► The study reports only the barriers experienced by parents.

decisions to reduce the child's suffering in the context of palliative care may also limit the life-expectancy of a child (such as discontinuing or reducing life-sustaining technology).[3 4]

Life-threatening conditions are defined as conditions for which curative treatment may be feasible, but for which this treatment could also fail, leading to a possibility of dying.[5] Care for children with life-threatening conditions can include aspects of both curative care and palliative care at the same time.[2] Healthcare providers are not always confident about their skills in providing palliative care.[6] The main reported barriers in paediatric palliative care as experienced by healthcare providers are communication, the uncertainty of the prognosis, time constraints and lack of education.[7–11] Studies on barriers experienced by parents show that communication is a major barrier for them.[12–17] Other frequently mentioned barriers are a lack of care for the family[12 13] and bereavement care.[17]

But knowledge of parents' experiences of barriers to good care is limited in at least two ways. First, it focusses mainly on specific health conditions, particularly on oncology,[14 15] and

## INTRODUCTION

Parents and physicians caring for children with life-threatening conditions face multiple challenges in providing the best care for children. Care for children whose futures and chances of survival are uncertain is often a complex trajectory, where curative care and palliative care entwines.[1 2] Curative care goals—to cure/prolong life—and, palliative care goals—to relieve suffering—sometimes collide, complicating care. Care for these children is further complicated because some

second, it focusses mainly on children receiving palliative care, often with a certain, lethal prognosis.[12 13 16 17] However, not all children who have life-threatening conditions receive palliative care, or are diagnosed with a lethal prognosis.[18–20]

To provide insight into barriers to care and decision-making for children living with life-threatening conditions, we conducted a qualitative study throughout the Netherlands. We interviewed parents who had experience in caring for a child with a life-threatening condition to identify what they regarded as barriers to both good care and good decision-making.

## METHODS

In a large-scale, nationwide qualitative study we interviewed parents of children, either living with a life-threatening or terminal condition, or who had died after a medical trajectory (a maximum of 5 years before the date of the interview). Children were aged between 1 and 12 years old and had a variety of life-threatening conditions.

## RECRUITMENT

Participants were recruited between November 2016 and October 2018. To recruit participants, all Dutch paediatricians received an invitation from the Paediatric Association of the Netherlands (NVK) to invite potential parents to join the research.[21] A website with information about the study was created to inform potential participants.[22] Parent support groups in oncology, cardiology, metabolic and neuromuscular diseases and palliative care helped to reach potential participants through their online platforms. To ensure maximum variety in ethnic background we recruited through physicians with expertise in treating patients with different cultural backgrounds. Sampling was aimed at obtaining maximum variety in terms of health condition, age, cultural background, parental level of education and place of residence. Three participants dropped out due to scheduling difficulties. Recruitment continued until thematic saturation and saturation of variety in participants was achieved.

## INTERVIEWS

Parents were interviewed in face-to-face in-depth interviews, usually held at the parent's place of residence. The first author (MA, PhD student, female) conducted the interviews after following formal training. The participants signed an informed consent form before the interview. Parents did not know the interviewer before the study. All participants were given the choice to conduct the interview alone or together with a co-parent. A topic guide had been developed as an interview guide. The interview covered the following themes: course of the disease, symptoms, suffering, care and decision-making and end-of-life. The topic guide is added as a online supplementary file to

the manuscript. The interviews were audio recorded and subsequently transcribed verbatim. All interviews were anonymised during transcription. The average duration of the interviews was 2 hours. Participants were given the opportunity to revise their transcribed interviews.

## ANALYSIS

For this article, we selected quotes concerning perceived barriers in care. For reasons of practicality in handling the approximately 2500 pages of transcripts, the first author gathered all codes on barriers in care in parent interviews, which were subsequently read by all authors to become familiarised with the content. The first author further focussed on parents' perceptions of barriers in care in the selection of interview fragments, and this was reviewed by all authors. The research team (the authors of this paper) consisted of a PhD student, an ethicist, a professor in end-of-life care and a child psychiatrist and a physician/professor in paediatric palliative care.

A primary thematic analysis, aiming at a qualitative description of barriers as perceived by parents,[23] was performed using the constant comparative approach.[24] Coding was performed by the first two authors and reviewed by all authors. Differences were settled by discussion until consensus was achieved.

## PATIENT AND PUBLIC INVOLVEMENT

The focus of this study is on parent experiences and preferences. During the study, biannual meetings were held with an advisory board (nine members) of parents, physicians and researchers who provided feedback on the findings. They were offered remuneration for their efforts. They provided substantial input on the study design, recruitment, analysis and the reporting of results of the study. All participants were given the opportunity to check and revise their manuscript, and are regularly updated on the outcomes of the study by email.

## RESULTS
### Participants

We interviewed 64 parents of 44 children. Our participants came from all over the Netherlands. All children received, or had received, care in at least one university medical centre, often combined with care in one or more local hospitals. All eight university hospitals in the Netherlands, as well as over 20 local hospitals, were represented in the study. Characteristics of children and parents are shown in tables 1 and 2 and in figure 1.

### Experienced barriers

We identified six major barriers in care for children with life-threatening conditions: We summarised the barriers in box 1.

| Table 1 | Characteristics of children (n=44) |
|---|---|
| Status, deceased | 27 (61.4%) |
| Gender, male | 20 (45.5%) |
| Age (years) | |
| 1–3 | 15 (34.0%) |
| 4–6 | 8 (18.2%) |
| 7–9 | 9 (20.5%) |
| 10–12 | 12 (27.3%) |
| Diagnosis | |
| Malignancies | 18 (40.9%) |
| Neurological/metabolic | 17 (38.6%) |
| Cardiovascular | 4 (9.1%) |
| Central nervous system | 3 (6.8%) |
| Other | 1 (2.2%) |
| Undiagnosed | 1 (2.2%) |

| Table 2 | Characteristics of parents (n=64 interviewed parents, 42 couples) |
|---|---|
| **Relationship status** | |
| Married/together | 34 (81.0%) |
| Level of education, mothers (n=42)* | |
| Low-level education (<4 years) | 1 (2.4%) |
| Mid-level education (practical, 4 years) | 15 (35.7%) |
| Higher education (vocational, 4 years), | 14 (33.3%) |
| Academic education (university, 4–6 years) | 12 (28.6%) |
| Level of education, fathers (n=42)* | |
| Low-level education (<4 years) | 4 (9.5%) |
| Mid-level education (practical, 4 years) | 15 (35.7%) |
| Higher education (vocational, 4 years), | 14 (33.3%) |
| Academic education (university, 4–6 years) | 9 (21.4%) |
| Ethnicity of parents, according to the participants (n=42) | |
| Dutch | 36 (85.7%) |
| Mixed | 6 (14.3%) |
| Religious/spiritual beliefs (n=64) | |
| None | 39 (60.9%) |
| Christian | 19 (29.7%) |
| Other | 6 (9.4%) |
| Family composition | |
| One child | 8 (19.0%) |
| Two children | 22 (52.4%) |
| Three children | 10 (23.8%) |
| Four or more children | 2 (4.8%) |

*We considered the educational level of both parents to be possibly of influence on the care and decisions of the child. Therefore the educational level of both parents is represented, even if one of them was not interviewed.

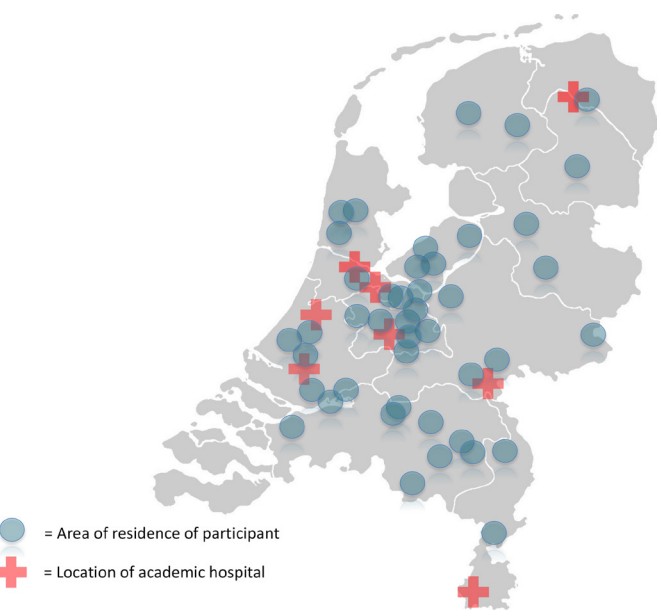

= Area of residence of participant

= Location of academic hospital

**Figure 1** Participants' places of residence.

## BARRIER 1: PARENTS WISHED FOR MORE EMPATHETIC AND OPEN COMMUNICATION ABOUT THE ILLNESS AND PROGNOSIS

Parents univocally felt that the wish to receive complete and open information, and the absence thereof, was a significant barrier to good care. Most experiences regarding communication involved either a perceived lack of openness about the prognosis or a lack of empathy in communication.

| Box 1 | Barriers in paediatric palliative care, as experienced by parents |
|---|---|

Barrier 1: Parents wished for more empathic and open communication about the illness and prognosis.
a. Absence of conversations about the future and possible prognosis.
b. Communication lacks empathy.
Barrier 2: Organisational barriers create bureaucratic obstacles and a lack of continuity of care.
a. Lack of continuity in care threatens good care
b. Healthcare structures create bureaucratic obstacles.
Barrier 3: Parents wished for more involvement in decision-making.
Barrier 4: Parents wished they had more support from the healthcare team on end-of-life decision-making.
Barrier 5: Parents experienced a lack of attention for the family during the illness and after the death of their child.
a. Impact on parents and siblings is neglected in care.
b. Parents experience a lack of bereavement care.
Barrier 6: Parents experience an overemphasis on symptom-treatment and lack of attention for their child as a person.
a. Lack of attention for how a child might experience treatment and illness.
b. Failure to evaluate symptoms in the context of the child's illness as a whole.
c. Overemphasising the patients role as a sick person instead of a human being.

### Absence of conversations about the future and possible prognosis

Telling a parent that their child may not survive might seem like a message that parents would not like to receive, but parents (both bereaved parents and non-bereaved parents) stressed that they had wished for open conversations about their child's future. Not talking about the prognosis creates uncertainty and makes the topic a taboo. They underlined the importance to openly talk about this, especially when the prognosis is uncertain.

> (F02): 'In the first few years, you still have the hope that things might change, but after a few years, you realise that almost all of that hope has gone. It would be great if physicians would take you by the hand and say: "We have to accept that this is what it is, and that things are probably not going to get better." And ask you as a parent: "What are your thoughts on that? Would you, despite that prognosis, like to keep on trying new treatments? Or do you think it is better not to do that?"'

Parents specifically needed openness about the prognosis to discuss the possibility of transition between different levels of concurrent life-prolonging treatments and comfort care, or to make end-of-life decisions. In a few cases, the topic of dying was only mentioned a day before the death of their child.

### Communication lacks empathy

Parents experienced a lack of empathetic communication. They felt that the friction between healthcare professionals doing their daily jobs and families going through a once-in-a-lifetime event was characteristic for the lack of sympathy that they experienced.

> (M27): '(the lady who came to explain the procedure of palliative sedation) was so insensitive. She just explained all the technicalities, with a broad smile. And I remember thinking: "It's not like I am buying bread from you!" (…) It was as if she thought: "let's get this over with", as if there was not an entire life, an entire family, an entire world affected by the procedure that she was explaining.'

But experienced barriers in communication were not limited to interactions with healthcare providers or hospital settings.

> (F16): 'Before we had even held the funeral, we received a letter from the local authority asking us to return our special parking permits as soon as possible. Just like that, without any condolences. Bizarre.'

### Barrier 2: organisational barriers create bureaucratic obstacles and a lack of continuity of care

With regard to organisation, two themes emerged: continuity of care and bureaucracy.

### Lack of continuity in care threatens quality

Parents described having to deal with many different care providers, either individual professionals or organisations, which made it difficult to guarantee continuity of care. Agreements made with one provider did not always dovetail well with the care that they received from other care providers.

> (F03): 'There was a physician on duty, and when we updated him on the 'do not resuscitate' agreements that we had made with our regular physician, he said: "No, I'm not going to do that. It is my shift, and my responsibility, and I'm not going to do that." You take all this time to make decisions with your doctor, and then another doctor, who happens to be on duty and doesn't know your child or the file, who has never had a conversation with you, simply states: "I'm responsible, so I determine what happens."'

### Healthcare structures create bureaucratic obstacles

Parents particularly experienced bureaucratic barriers to good care in situations where they had to deal with a multitude of healthcare-related organisations.

> (M31): 'When the neurologist prescribes some sort of medicine, he has to send it by fax to the pharmacist, he can't send it by email. And if there is even one comma in the wrong place, the pharmacist won't prescribe the medicine that my son needs.'

Several parents expressed the need for a case manager, who could bridge the gaps between parents, different healthcare providers and financers. Notably, this barrier was also mentioned by parents who had received palliative care from teams that already worked with case managers. While more and more paediatric palliative teams in the Netherlands are appointing case managers to families dealing with a life-threatening illness, the question arises as to how they can be used in the most effective manner.

### Barrier 3: parents wished for more involvement in decision-making

The third barrier concerned participation in decision-making. There was a lot of variation regarding the involvement of parents in decision-making. Parents who had been involved in decision-making, appreciated it.

> (M15): '(when we had to decide whether to go for radiotherapy or not) our doctor said: "Whatever your decision is, there is no wrong answer, and I will stand by your choice."'

Other parents described that they were not as involved in decision-making as they would have liked, both for decisions made during curative treatment and for those made in the terminal phase. One parent described that the decision to withdraw life-sustaining treatment for their child was made by a neurologist who had never spoken to them.

> (M27): 'The neurologist never spoke to us about his motivations to (withdraw treatment) and start the death of our son. He came in, made the observation

right in front of us, but only spoke in private to our paediatrician, and left.'

All participants stressed the importance of involving parents in the decision-making process, stating that the decisions included considerations about the suffering, quality of life and daily life of the child, something in which the parents' expertise was needed. The only exception parents mentioned to being involved, were decisions that parents saw as 'merely medical' information, such as technicalities about medication or treatment.

### Barrier 4: parents wished they had more support from the healthcare team at end-of-life decision-making

Parents mentioned discussions about the possibility of transition between different levels of concurrent life-prolonging treatments and comfort care, or to make end-of-life decisions, as major themes in the care for their child. Parents sometimes had doubts about the appropriateness of continued treatment or felt that it was wrong that parents themselves had to initiate a conversation about end-of-life decision-making.

Additionally, several parents expressed a wish for more legal possibilities for active life-ending in order to relieve their child's suffering and grant their child a dignified end of life. In the Netherlands, active life-ending for children—sometimes referred to as paediatric euthanasia—is not legal for children aged between 1 and 12 years old.[25] Withholding or withdrawing treatment and palliative sedation are considered permitted medical practices.[4]

For the majority of the children in the study, end-of-life decisions were made, ranging from withholding or withdrawing life-prolonging or life-sustaining treatment to palliative sedation.

Nevertheless, in several cases, parents still felt the need for an option to make end-of-life decisions. In most cases, their need came from a situation in which the condition of their child was physiologically stable and the physicians could not justify the withdrawal of treatment or palliative sedation.

(M02): 'Physicians told us that if they had to (give her) so much morphine to ease her pain that the morphine itself would be the cause of her death, it would be considered active life-ending, which is not allowed for children who cannot ask for it themselves. But we would want that for her, because we have the feeling that the only thing that can bring her comfort and peace (emotional) will be death.'

In other situations, parents felt a need for active life-ending because the existing options were unable to grant their child a dignified end of life.

(M07): 'If we all know that the palliative sedation is going to result in his death, why does he have to go through these final days of suffering? Why does it have to be so… undignified?'

### Barrier 5: parents experienced a lack of attention for the family during the illness and after the death of their child

A life-threatening illness does not only affect children, but also their families. Parents felt that they and their families were not always supported during and after the illness of their child. Siblings in particular were often overlooked.

(M20): 'As a parent you get a little support, but for brothers and sisters there is very little to absolutely no help at all. They have to figure it out for themselves, it seems as though they are forgotten.'

While support for the family is one of the aims of paediatric palliative care,[4] and all Dutch hospitals have psychologists, social workers and/or palliative care teams to provide this, it seems that this support is not always offered to parents or did not function as well as parents had hoped. However, for parents, the need was not so much about offering psychological support as it was about empathy in their contact with the healthcare providers that they came across.

(M05): 'What they don't always realise is that the hospital also becomes your social world; the contact with doctors, the chats with nurses.'

### Barrier 6: parents experience an overemphasis on symptom-treatment and lack of attention for their child as a person

The final barrier expressed by parents was a wish for healthcare providers to understand the fact that patients are persons, not merely carriers of certain symptoms. They explained that healthcare professionals often failed to evaluate the impact of the condition or the medical interventions on the child's daily life. This barrier had three different aspects: (1) a lack of attention for how a child might understand and cope with treatment and illness, (2) failure to see the complexity of the child as a whole and, finally, (3) overemphasising the child as a patient instead of as a whole person.

### Lack of attention for how a child might experience treatment and illness

One specific characteristic of paediatric care is the attention given to how children understand and cope with their condition and the associated treatment. Parents underlined that this is important both for 'bigger questions' and for simple medical interventions. Parents mentioned the comprehension of the child as a vital aspect in the child's ability to cope with illness and treatment.

(M04): 'Hospitals are busy with production. So, a nurse comes in, and basically jumps on the child to take his blood pressure, or give an injection, without paying attention to what it does to him (…) And I often wonder: How hard can it be to ask yourself beforehand how to approach this individual child, so that he understands the interventions?'

Parents saw taking the time to explain to children what is happening to them (instead of performing treatment

on them without explanation) as an important aspect of taking children seriously as people.

### Failure to evaluate symptoms in the context of the child's illness as a whole

Children with life-threatening conditions can receive treatment from various specialists. Parents described how specialists sometimes failed to look beyond the specific symptoms that they were treating and see the child and the complexity of their condition as a whole.

> (M29): 'We had an incident because my son had fallen, and the dental surgeon took one look at his teeth and said: "Those teeth have got to go." And I said: "But what then? He already has problems swallowing." And he said casually: "He'll need a feeding tube." Not caring what it would mean to him, and our lives, to make that decision. And I don't think he even saw the boy who was sitting there, who I love so much and who I am so worried about.'

### Overemphasising the patients role as a sick person instead of a human being

The final barrier identified was the overemphasis of the child as a patient, at the cost of the child as a person. Parents described how medical treatment sometimes overemphasised the child as a patient and that there was little attention for the child as a person, impeding the child's quality of life. Parents stressed that especially for ill children keeping up a normal life became very important: going to school, not standing out and feeling like an ordinary child. Daily activities that did not focus on the child's illness, but on the child as an ordinary person, became cherished. As this mother remembers:

> (M14): 'It was the highlight of his day, when his teacher would come; solving math problems (laughs).'

This barrier also influenced the readiness of a child to communicate about their illness. Children did not always want to talk about the illness, a few even forbidding their parents to talk about it with others. Seeing the child too much as a patient, either by focussing merely on the child's illness, or by trying to make them feel special, might deprive children of living a normal life.

Both barriers 5 and 6 refer to what parents described as their wish for a more human-centred approach to care, which also implied an empathic connection between professionals and parents/patients. According to the parents, some physicians perceived this attitude as unprofessional:

> (M20): 'One physician had a big impact on me. We had to go home, and she said goodbye to my son. She started crying and apologised: "I'm so sorry, I'm being very unprofessional."'

For parents, however, this kind of connection was much appreciated and in fact seen as a sign of professionalism.

(M20, continued): 'And I said: "No, right now, you are human." I wonder if she knows that of all those doctors, she is the only one that crosses my mind frequently. Because she cared.'

## DISCUSSION
### Main findings

This qualitative study gives insight into parents' perceived barriers in care for children with life-threatening conditions. The barriers experienced by parents focussed almost without exception on the non-technical skills in medicine: attention for children, empathy, communication, decision-making and organisation. The absence of perceived barriers on the execution of symptomatic care is notable.

With respect to the first barrier, communication, other studies report communication as a barrier for both physicians[7 8 11 26–30] and parents.[12–16 31–33] Our results underline the importance of openness as a requirement for decision-making, and adds that communication is not limited to hospital settings. A possible future development for paediatric palliative care is to extend communication training to all professionals dealing with children with life-threatening conditions, not merely hospital workers.

To reduce organisational barriers, parents suggested introducing case managers that might help parents and children with bridging the gaps between different healthcare professionals and in dealing with bureaucratic difficulties. Over the past year, several paediatric palliative care teams in the Netherlands have introduced such case managers, with the aim of trying to reduce this barrier. However, it is worth mentioning that in current practice, palliative care is predominantly limited to terminally ill children, and consequently, these palliative care services also remain limited to the terminal phase. However, issues with the continuity of care and with bureaucracy may arise long before this phase.

With regard to the third barrier, decision-making, the involvement of parents in decision-making varied. Previous studies also show that parents are not always involved in decision-making.[34 35] However, while these studies illustrate that some parents do not wish to be involved in decision-making,[36 37] all parents in our study expressed the importance of being involved in decision-making, with the only exception of decisions that involved merely medical knowledge. These differences may be explained by the different timing of decision-making processes (critical care vs longer-term care). It is also possible that the understanding of parents on what it means to be involved differs between studies (for example, being heard vs being fully responsible).

Shared decision-making has been an important movement in both paediatrics and medicine in general over the last decade.[38 39] Dutch regulations underline the importance of making parents, and—if possible—even children themselves, part of the decision-making process.[4] The fact

that parents often feel excluded may be an incentive to further investigate why parents are not always included, and what is needed to achieve this.

The perceived need for options for active end-of-life decisions (eg, euthanasia or active life-ending on request of parents) is a barrier that, to our knowledge, has not previously been mentioned in studies on barriers in care for children with life-threatening conditions. In contrast, other studies mention parents' discomfort with making end-of-life decisions such as withholding or withdrawing treatment.[40] Although the Netherlands is known for having a legal regulation of euthanasia and end-of-life decision-making in adults and newborns,[25 41 42] the current regulations for children aged between 1 and 12 years old are similar to those in most other countries. Withholding and withdrawing treatment and palliative sedation are permitted, but active life-ending (euthanasia) is not. Following the Belgian legalisation of euthanasia for competent minors, there has been a debate in the Netherlands about whether regulations on euthanasia should be broadened to allow active life-ending for children younger than 12 in cases of unbearable and hopeless suffering.[43 44]

Parents perceived a lack of care for the family. Family care is one of the goals of palliative care[8] and is arguably a service that all parents and siblings of the child should receive.[18] Previous studies highlight the importance of family support, including bereavement care.[12 13 17]

Previous studies have underlined that care for young children includes attention for a child's development.[18 45] Our study adds that this does not only mean seeing the child as a whole, with both physical, social and emotional aspects, but that it also means seeing the child as a *person*. From the perspective of parents, care for children with life-threatening conditions focusses too much on the symptoms and not enough on the human beings behind the symptoms. This raises the question of whether such an attitude should be a part of medical professionalism.

### Medical professionalism: dehumanisation?

Medicine and emotional intimacy seem to have a difficult relationship. Can a professional working in paediatric palliative medicine be expected to build an empathic, intimate relationship with all of their patients? Some authors argue that what makes a good doctor is, in essence, expertise and knowledge, not empathy; that good doctor may be stand-offish or surly, as long as they treat a symptom adequately.[46] Intimacy is a tool needed to detect the needs of a patient, but should be used very carefully, otherwise a doctor might 'lose himself'.[46–48] Nevertheless, empathy benefits patients.[49] The view of detachedness as a professional quality is often—although not always explicitly—reiterated throughout medicine. There is a tension between this viewpoint of professional detachedness and that of empathic intimacy, or human-to-human contact, that parents ask for.

What parents seem to miss in current practise is the other connotation of the word 'care': to care about. We

cannot expect physicians or nurses to become emotionally attached to every patient that they meet, but this is not what parents seem to miss in current care. What they miss in care is not the lack of showing emotion, but the affirmation that all parties—children, family and physicians—are in fact human and that the illness affects lives, not monitors. This attitude can be implemented into standard care for children with life-threatening conditions.

Medical detachedness versus attention for persons is not black or white, but rather a continuum, ranging from patient-physician relationships that focus on very specific symptoms, without much attention for the person who is experiencing it, towards recognising the complexity of the child's overall health situation and finally evaluating care in the context of a child who as a human being experiences its illness. In different forms of medicine different levels of detachedness, and attention for the person may be appropriate. A dental appointment, where the physician-patient relationship is short, and the treatment limited to a specific expertise, calls for a different kind of professionalism than paediatric palliative care, where its very goal implies attention for the person. By its goal—improving quality of life by relieving suffering[2]—paediatric palliative care has shifted its focus from the illness and symptom towards the patient. The experienced barriers indicate that this shift is possibly insufficiently translated into care. Perhaps it is time to consider how we should evaluate professional detachedness in the light of the goals of palliative care. Above all, policies and guidelines should not merely guide professionals on the technical aspects of symptom treatment, but aid them in topics of professional intimacy as well.

### Strengths and limitations

Our study has some limitations. First, our study focussed on young children (aged 1 to 12 years old) and the results cannot be generalised to adolescents or neonates. Second, although we put considerable effort into recruiting participants with different cultural and religious backgrounds, the success in achieving this variation was limited. Finally, we did not include the perspectives of physicians, in order to solely present the barriers perceived by parents. Although the parents' own experiences are the experiences that they live by, and that represent their reality, they do not present a complete overview alone. In order to fully evaluate care for children with life-threatening conditions, the reality of the physicians, nurses and other involved parties should also be described.

The strengths of this study are the size of the study. We held in-depth interviews with 64 participating parents, and over 110 hours of qualitative data on audio were collected. This makes the study unique in its size. A second strength is the variety. Most existing research is limited either by medical diagnosis or hospital where care is administered. In this study we interviewed parents from children with various illnesses, in various stages of their illness, receiving care from various hospitals, providing

a much needed broad perspective on care given to children living with life-threatening conditions.

## CONCLUSION

This study explores the barriers that parents encounter during the process of caring for children with life-threatening conditions. The barriers experienced by parents in care for children with life-threatening conditions and uncertain futures are related to six major themes. First, parents wished for more empathetic and open communication about the illness and prognosis. Second, organisational barriers create bureaucratic obstacles and a lack of continuity of care. Third, parents wished for more involvement in decision-making. Fourth, parents wished they had more support from the healthcare team on end-of-life decision-making. Fifth, parents experienced a lack of attention for the family during the illness and after the death of their child. Finally, parents experience an overemphasis on symptom-treatment and lack of attention for their child as a person. The parents' wish to see the child as a person sheds new light on the relationship between medical professionalism and detachedness. We argue that in paediatric palliative care, the child behind the symptoms is sometimes forgotten. Paediatric palliative care might suit the needs of parents and children better when it re-evaluates its current professional detachedness, and progresses towards a medical professionalism where not only symptoms, but also people are treated.

**Contributors** All authors are responsible for the reported research and article and meet the ICMJE standards for authorship. All authors agree to be accountable for all aspects of the work in ensuring that questions related to the accuracy or integrity of any part of the work are appropriately investigated and resolved. MB contributed to the study design, conducted the recruitment and interviews, performed the qualitative analysis and drafted the manuscript. EM contributed to the study design, helped in recruiting participants, performed the qualitative analysis and contributed to the writing of the manuscript. AvdH contributed to the study design, checked the qualitative analysis and contributed to the writing of the manuscript. IH contributed to the study design, contributed to the study design, checked the qualitative analysis and contributed to the writing of the manuscript. EV contributed to the study design, helped in recruiting participants, checked the qualitative analysis and contributed to the writing of the manuscript.

**Funding** This work was supported by 'Ministerie van Volksgezondheid, Welzijn en Sport'. Grant number 324717 (560019968).

**Map disclaimer** The depiction of boundaries on this map does not imply the expression of any opinion whatsoever on the part of BMJ (or any member of its group) concerning the legal status of any country, territory, jurisdiction or area or of its authorities. This map is provided without any warranty of any kind, either expressed or implied.

**Competing interests** None declared.

**Patient and public involvement** Patients and/or the public were involved in the design, or conduct, or reporting, or dissemination plans of this research. Refer to the Methods section for further details.

**Patient consent for publication** Not required.

**Ethics approval** The study was reviewed and deemed exempt by the medical ethical committee of the University Medical Center Groningen. Number: METC2016.223.

**Provenance and peer review** Not commissioned; externally peer reviewed.

**Data availability statement** No data are available. There is no additional data available.

**ORCID iD**
Marije Brouwer http://orcid.org/0000-0002-5464-355X

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
