## [Reviewer comments · BMJ Open]

ARTICLE DETAILS

TITLE (PROVISIONAL)	Barriers in Care for Children With Life-Threatening Conditions: a Qualitative Interview Study in the Netherlands
AUTHORS	Brouwer, Marije; Maeckelberghe, Els; van der Heide, Agnes; Hein, Irma; Verhagen, Eduard

VERSION 1 - REVIEW

REVIEWER	Douglas L. Hill The Children's Hospital of Philadelphia, USA
REVIEW RETURNED	21-Jan-2020

GENERAL COMMENTS	Review of "They are Persons, not Symptoms: Parents Express Barriers in Care for Children With Life-Threatening Conditions" (Manuscript ID bmjopen-2019-035863) This is a qualitative study of 64 parents of 44 children with life threatening conditions in the Netherlands. The first author conducted in-person interviews that included parent perceptions of perceived barriers to care. The authors identified 6 barriers to care mentioned by parents in the interviews: 1) communication, 2) a lack of continuity in care, 3) a lack of shared decision-making, 4) end-of-life decision-making, 5) a lack of attention for the family's welfare, 6) failing to see the child as a person. The authors conclude that barriers identified by parents focused almost exclusively on non-medical aspects of care for children with a life-threatening condition suggesting that current care focuses too much on symptoms and not enough on the human beings behind those symptoms. The authors highlight strengths of the study including the large sample and the inclusion of a wide variety of medical conditions in various stage of illness (e.g. not just pediatric oncology patients or pediatric patients at the end of life). This is an interesting paper that clearly describes some of the key sources of distress and frustration for parents of children with life threatening illness. While many of the findings are familiar from earlier research, the current study shows these issues are prevalent across the range of life-threatening illnesses, not just cancer or end of life. The authors report findings of a desire on the part of some parents for more active forms of euthanasia for children this age instead of the current options in the Netherlands of palliative sedation and withdrawal of treatment. The authors may, however, wish to strengthen their paper by making some changes and clarifications. Overall there are some minor
--

English language issues involving awkward phrases that are unclear.

Abstract

Page 2, lines 24-38: While the core ideas are consistent, the six barriers to care are described differently in different sections of the paper. The authors may wish to decide what the overall label is for each barrier, and what the subtopics are under each barrier and refer to these consistently throughout. Table 4 was the clearest presentation of the barriers and subtopics.

What is already known about this topic

Page 4, Lines 5-7: The following sentence doesn't quite make sense:

"Healthcare providers describe as the main barriers is palliative care: communication, an uncertain prognosis, time constraints and lack of education"

perhaps:

"Healthcare providers describe the main barriers in palliative care as communication, an uncertain prognosis, time constraints and lack of education about palliative care."

What this study adds

Page 4 Lines 24-25: The authors state:

"The parents' wish to see the child as a person sheds new light on the relationship between medical professionalism and detachment."

The authors may want to clarify that they are referring to the parents wish for health providers to see the child as a person. The authors may also wish to clarify the problem they are referring to by describing it as "the conflict between medical professionalism and empathy."

Introduction

Page 5, lines 16-17: The authors refer several times in the paper to "decisions about the proportionality of life-prolonging treatments and sometimes end-of-life decisions."

It's not quite clear what this means. The authors may be referring to decisions about transitioning between different levels of concurrent life-prolonging treatments and comfort care.

Methods

Page 6, lines 33-45: The authors do not say much about the content of the interview, what topics were covered in the interviews (which were an average of two hours), or provide any examples of questions asked. Were all of the questions about barriers to care, or were other topics covered? If a script or list of questions was used, the authors may wish to provide this as a supplementary file.

Page 6, lines 51-54: "the first author gather all codes on communication in parent interviews".

Does this mean the first author went through the transcripts and

identified all sections related to communication for the other authors to review?

Page 7, line 12: "Families of patients" instead of "Family of patients"

Results

Page 8, table 2: What does low, mid, high level of education mean? Perhaps give a range of years of education completed for each category or find some other way to make these categories more meaningful.

Page 9-15: As mentioned for the abstract, the categories and sub-categories of barriers are described differently in different parts of the paper. The list on Page 9 lines 31-37 doesn't match the headings of the rest of the results or table 4.

Table 4 should be moved to the beginning of the results section.

Page 10, lines 14-16: again the unclear phrase "proportionality of treatment"

Page 11 line 43 – Page 12 line 12: The authors provide good examples of parents being satisfied that they were involved in important decisions and being unsatisfied that they were excluded from important decisions. However, the authors may wish to say more here and in the discussion about whether there was any variation in how much the parents wanted to be involved in the decisions. Prior research has found that parents range from wanting to make important decisions all by themselves, wanting to share them with the medical team, and wanting the medical team to make the important decisions. In this paper, the default assumption seems to be that all parents wanted to be involved in the important decisions.

Madrigal VN, Carroll KW, Hexem KR, Faerber JA, Morrison WE, Feudtner C. Parental decision-making preferences in the pediatric intensive care unit. *Crit Care Med.* 2012;40(10):2876-2882.

Page 12 Lines 18-22: "A significant number of parents had doubts about the appropriateness of continued treatment or felt that it was wrong that parents themselves had to initiate a conversation about end-of-life decision-making."

The authors may wish to highlight this finding in the discussion. Health care providers often avoid bringing up palliative care or end-of-life decisions because they think families are not ready, when in fact some families are relieved when the health care provider finally opens this discussion because they didn't want to bring it up themselves.

Page 13 line 42 – Page 15 line 29: The authors report several important and interesting findings in this section which overall is about the importance of treating the child as a person not as a set of symptoms. However, the sub-topics and labels are vague and confusing.

The first subtopic "Child's understanding of the illness" needs a little more explanation to fit under the topic of treating the child as a person (e.g. taking the time to explain what is happening is part of how you show respect to a person, versus just coming in and

	treating them). The second subtopic (described as either “the child and the illness” or “failure to see the complexity of the child as a whole” seems to be more about specialists focusing on one aspect of the child’s health while ignoring the larger complicated picture (for example a nephrologist saying things are going well because kidney function is fine when in fact the child is declining because of other organ systems failing). The authors may wish to distinguish the subtopic more clearly from the last subtopic which seems to be more about treating the child as a person not as a collection of symptoms. Discussion As noted above, Table 4 would make more sense in the results. Table 4 (other than 6b) is also the clearest outline of the barrier categories and sub-categories). Page 16 line 58 – Page 17 Line 11: As noted above, the authors may wish to discuss the fact that other studies have found a subcategory of parents that do not want to be involved in decision making and whether there was any evidence of this in this sample. Page 17 line 38: Perhaps replace “a facility that all parents . . . should receive” with “a service that all parents . . . should receive”. Page 17 lines 47-50: The authors state “Our study adds that this does not only mean seeing the child as a whole, with both physical, social and emotional aspects, but that it also means seeing the child as a person.” The distinction between seeing the child as a whole (including psychical, social, and emotional aspects) and seeing the child as a person is unclear. The authors may want to present this on a continuum with one extreme being focusing on one symptom/body part (the dentist and the teeth), the middle being recognizing the complexity of the child’s overall health situation but still being very detached and clinical (e.g. treating the child as a complex set of symptoms), and the other extreme of seeing the child as a living person who is experiencing all these things. Page 19, line 14: The paper ends suddenly after the strengths and limitations. The authors may wish to add a final section on overall implications and conclusions.
--	--

REVIEWER	Marjorie S Rosenthal Yale University, United States
REVIEW RETURNED	30-Jan-2020

GENERAL COMMENTS	ABSTRACT AND INTRO: well written and clear METHODS: please expand upon your description of methods. Who wrote the interview guide? What was their expertise? Was it different for different diseases or different respondents? What was the interview guide based on/adapted from? Who was on the analysis team? What was their expertise? Was there an opportunity to share the themes with the parents who were
--

	interviewed? How many people were on the advisory board? Did people get paid to participate in it? How did the authors settle on the sample size they did? Was there any rationale? RESULTS: Overall the results were clear and the headings leading to the quotations were clear. What the authors should do to make this section much more informative would be for them to declare a theme--not just stating that communication, decision-making etc is a barrier but be more specific in the heading--tell the reader your analysis of what the actual issue is. Otherwise, the analysis appears to be superficial--all you did was group the quotations by topic, you didn't try to get to the themes of the quotations. For example, something like "Parents wished they had more support from the health care team on end-of-life decision making" is more informative than "End-of-life decision-making" The authors should do this for all 6 themes and make Table 4 reflect this as well. The authors use quantitative language in the results section for example, "a majority of the parents," When, in fact, that's inappropriate to do in qualitative research--it is mis-leading to the reader bc the sample was not created for that purpose--it wasn't created for quantitative reasons or through quantitative means. Is there a Table 3? TABLE 2 The n for maternal education should be n=21. Same for paternal Small points: Page 2, line 7--should be "barriers, as perceived by parent TO good care..." Page 4, lines 5-7--this sentence is unclear: "Healthcare providers describe as the main barriers is palliative care: communication, an uncertain prognosis, time constraints and lack of education" Page 6, line 43--"Participants were given the opportunity to revise their manuscripts."this sentence is unclear. What do you mean by "manuscript"? Page 11, line 33--What is a "subsidy provider"?
--	--

VERSION 1 – AUTHOR RESPONSE

Reviewer: 1 Reviewer Name Douglas L. Hill	
This is an interesting paper that clearly describes some of the key sources of distress and frustration for parents of children with life threatening illness.	We thank the reviewer for these kind words. We will address the changes and clarifications in more detail below.

While many of the findings are familiar from earlier research, the current study shows these issues are prevalent across the range of life-threatening illnesses, not just cancer or end of life. The authors report findings of a desire on the part of some parents for more active forms of euthanasia for children this age instead of the current options in the Netherlands of palliative sedation and withdrawal of treatment. The authors may, however, wish to strengthen their paper by making some changes and clarifications.	
Overall there are some minor English language issues involving awkward phrases that are unclear.	Thank you for bringing this to our attention. We have made changes throughout the text to improve clarity and readability of the manuscript.
Page 2, lines 24-38: While the core ideas are consistent, the six barriers to care are described differently in different sections of the paper. The authors may wish to decide what the overall label is for each barrier, and what the subtopics are under each barrier and refer to these consistently throughout. Table 4 was the clearest presentation of the barriers and subtopics.	Thank you for this important feedback, we have modified the text for more consistency. In line with the feedback of reviewer 2, we have rewritten the topics into themes.
What is already known about this topic Page 4, Lines 5-7: The following sentence doesn't quite make sense: "Healthcare providers describe as the main barriers is palliative care: communication, an uncertain prognosis, time constraints and lack of education" perhaps: "Healthcare providers describe the main barriers in palliative care as communication, an uncertain prognosis, time constraints and lack of education about palliative care."	Thank you for these suggestions. Upon editorial request, this section has been removed from the manuscript.
What this study adds Page 4 Lines 24-25: The authors state: "The parents' wish to see the child as a person sheds new light on the relationship between medical professionalism and detachedness." The authors may want to clarify that they are referring to the parents wish for health providers to see the child as a person. The authors may also wish to clarify the problem they are referring to by describing it as "the conflict between medical professionalism and empathy."	Upon editorial request, the section has been removed from the manuscript, but we moved the text in the conclusion-section, and used your suggestions there to improve the text in the conclusion-section. (see p.16)
Introduction Page 5, lines 16-17: The authors refer several times in the paper to "decisions about the proportionality of life-prolonging treatments and sometimes end-of-life decisions." It's not quite clear what this means.	Thank you for this helpful feedback. We have altered the text to provide more clarity. The new text is: "Care for these children is further complicated because decisions that may limit the life-expectancy of a child (including end-of-life decisions) are considered part of

The authors may be referring to decisions about transitioning between different levels of concurrent life-prolonging treatments and comfort care.	palliative care.” see p.4) p. 9: Parents specifically needed openness about the prognosis to discuss the possibility to transition between different levels of concurrent life-prolonging treatments and comfort care, or to make end-of-life decisions.
Methods Page 6, lines 33-45: The authors do not say much about the content of the interview, what topics were covered in the interviews (which were an average of two hours), or provide any examples of questions asked. Were all of the questions about barriers to care, or were other topics covered? If a script or list of questions was used, the authors may wish to provide this as a supplementary file.	During the interviews, a topic guide was used as an interview guide. We added information about the topic guide to the methods section, and included the topic guide as a supplementary file to the manuscript. The new text is: “A topic guide had been developed as an interview guide. The interview covered the following themes: course of the disease, aspects of the illness (symptoms and impact) suffering, care, decision-making and end of life. The topic guide is added as a supplementary file to the manuscript.” (p.5)
Page 6, lines 51-54: “the first author gather all codes on communication in parent interviews”. Does this mean the first author went through the transcripts and identified all sections related to communication for the other authors to review?	Thank you for spotting this inconsistency. We mistakenly wrote ‘communication’ instead of ‘barriers in care’. We have corrected the mistake. The new text is: the first author gathered all codes on barriers in care in parent interviews, which were subsequently read by all authors to become familiarized with the content. (see p.6)
Page 7, line 12: “Families of patients” instead of “Family of patients”	Thank you for this feedback, we have altered the text to match your suggestion
Page 8, table 2: What does low, mid, high level of education mean? Perhaps give a range of years of education completed for each category or find some other way to make these categories more meaningful.	Thank you for this useful suggestion. Given the educational system in the Netherlands, we decided to use the following distinction: Low level (less than 4 years) Mid-level (Practical, 4 years) high level (vocational, 4 years), and academic (university, 4-6 years)
Page 9-15: As mentioned for the abstract, the categories and sub-categories of barriers are described differently in different parts of the paper. The list on Page 9 lines 31-37 doesn’t match the headings of the rest of the results or table 4.	Thank you for this important feedback, we have modified the themes to provide consistency throughout the manuscript and the tables. In line with the feedback of reviewer 2, we have rewritten the topics into themes. The themes are now phrased as follows:  1) Parents wished for more empathic and open communication about the illness and prognosis. 2) Organizational barriers create bureaucratic obstacles and a lack of continuity of care. 3) Parents wished for more involvement in decision-making. 4) Parents wished they had more support from the health care team on end-of-life decision-making.

	5) Parents experienced a lack of attention for the family during the illness and after the death of their child. 6) Parents experienced an overemphasis on symptom-treatment and lack of attention for their child as a person. These descriptions are used throughout the manuscript to provide more consistency
Table 4 should be moved to the beginning of the results section.	We moved table 4 to the beginning of the results-section.
Page 10, lines 14-16: again the unclear phrase “proportionality of treatment”	We have modified the text for more clarity. The new text is: “For many participants, discussions about the possibility to transition between different levels of concurrent life-prolonging treatments and comfort care, or to make end-of-life decisions, were major themes in the care for their child.” (see p.11)
Page 11 line 43 – Page 12 line 12: The authors provide good examples of parents being satisfied that they were involved in important decisions and being unsatisfied that they were excluded from important decisions. However, the authors may wish to say more here and in the discussion about whether there was any variation in how much the parents wanted to be involved in the decisions. Prior research has found that parents range from wanting to make important decisions all by themselves, wanting to share them with the medical team, and wanting the medical team to make the important decisions. In this paper, the default assumption seems to be that all parents wanted to be involved in the important decisions. Madrigal VN, Carroll KW, Hexem KR, Faerber JA, Morrison WE, Feudtner C. Parental decisionmaking preferences in the pediatric intensive care unit. Crit Care Med. 2012;40(10):2876-2882.	Thank you for this useful suggestion. Indeed, compared to earlier studies, our parents were much less divided on their wish to be involved in decision-making. We have added a reflection on this finding in the results-section. The added text is: We also reflected on the finding in the discussion section. However, while in previous studies illustrate that some parents do not wish to be involved in decision-making,[37-38] all parents in our study expressed the importance of being involved in decision-making, with the only exception of decisions that involved merely medical knowledge. These differences may be explained by the different timing of decision-making processes (critical care versus longer term care). It is also possible that the understanding of parents on what it means to be involved differs between studies (for example, being heard versus being fully responsible). (see p.15) In the results-section, we added one nuance to the paper: In some cases parents reflected on decisions that they saw as merely medical decisions. The added text is: All participants stressed the importance of involving parents in the decision-making process, stating that the decisions included considerations about the suffering, quality of life and daily life of the child, something in which the parents’ expertise was needed. The only exception parents mentioned to being involved, were decisions that parents saw as ‘merely medical’ information, such as doses of medication or technicalities of treatment. (see p.11)

Page 12 Lines 18-22: “A significant number of parents had doubts about the appropriateness of continued treatment or felt that it was wrong that parents themselves had to initiate a conversation about end-of-life decision-making.” The authors may wish to highlight this finding in the discussion. Health care providers often avoid bringing up palliative care or end-of-life decisions because they think families are not ready, when in fact some families are relieved when the health care provider finally opens this discussion because they didn’t want to bring it up themselves.	Thank you for this suggestion. We agree with the reviewer on the importance of this finding, but the attention given to this finding is limited in this paper because we have written another paper on bad news communication, where this finding is further explored. This manuscript is currently being revised for another journal.
Page 13 line 42 – Page 15 line 29: The authors report several important and interesting findings in this section which overall is about the importance of treating the child as a person not as a set of symptoms. However, the sub-topics and labels are vague and confusing. The first subtopic “Child’s understanding of the illness” needs a little more explanation to fit under the topic of treating the child as a person (e.g. taking the time to explain what is happening is part of how you show respect to a person, versus just coming in and treating them). The second subtopic (described as either “the child and the illness” or “failure to see the complexity of the child as a whole” seems to be more about specialists focusing on one aspect of the child’s health while ignoring the larger complicated picture (for example a nephrologist saying things are going well because kidney function is fine when in fact the child is declining because of other organ systems failing). The authors may wish to distinguish the subtopic more clearly from the last subtopic which seems to be more about treating the child as a person not as a collection of symptoms.	Thank you for this important feedback. We have made several changes to improve the clarity of this finding. We have renamed the topic and subtopics for more clarity. The topics are now described as follows: 6. Parents experience an overemphasis on symptom-treatment and lack of attention for their child as a person. 6a. Lack of attention for how a child might understand treatment and illness. 6b. Failure to evaluate symptoms in the context of the child’s illness as a whole. 6c. Overemphasizing the patient instead of the person. - Along your suggestion, we added extra information to the first subtopic to explain why it fits under theme 6. The added tekst is: “Parents mentioned the understanding of the child as a vital aspect in the child’s ability to cope with illness and treatment.” [quote] Parents saw taking the time to explain to children what is happening them (instead of performing treatment on them without explanation) as an important aspect of taking children seriously as people. - We distinguished subtopic 6b and 6c from each other by clarifying the titles of the subtopics. The titles are now: 6b. Failure to evaluate symptoms in the context of the child’s illness as a whole. 6c. Overemphasizing the patient instead of the person.
As noted above, Table 4 would make more sense in the results. Table 4 (other than 6b) is also the clearest outline of the barrier categories and sub-categories).	Thank you for this suggestion. We have moved table 4 (now table 3) towards the results section. (see p.8)

Discussion Page 16 line 58 – Page 17 Line 11: As noted above, the authors may wish to discuss the fact that other studies have found a subcategory of parents that do not want to be involved in decision making and whether there was any evidence of this in this sample.	We have added this finding to the discussion-section. The added text is: However, while in previous studies show groups of parents that do not wish to be involved in decision-making, all parents in our study expressed the importance of being involved in decision-making, with the only exception of decisions that involved merely medical knowledge. These differences may be explained by the timing of decision-making processes (critical care versus longer term care) but it is also possible that there are cultural differences. It is also possible that the understanding of parents on what it means to be involved differs between studies (for example, being heard versus being fully responsible). (see p.15)
Page 17 line 38: Perhaps replace “a facility that all parents . . . should receive” with “a service that all parents . . . should receive”.	We have used your suggestion to improve the clarity of the text. The new text is: Family care is one of the goals of palliative care[8] and is arguably a service that all parents and siblings of the child should receive. (see p.15)
Page 17 lines 47-50: The authors state “Our study adds that this does not only mean seeing the child as a whole, with both physical, social and emotional aspects, but that it also means seeing the child as a person.” The distinction between seeing the child as a whole (including psychical, social, and emotional aspects) and seeing the child as a person is unclear. The authors may want to present this on a continuum with one extreme being focusing on one symptom/body part (the dentist and the teeth), the middle being recognizing the complexity of the child’s overall health situation but still being very detached and clinical (e.g. treating the child as a complex set of symptoms), and the other extreme of seeing the child as a living person who is experiencing all these things.	We thank the reviewer for this very helpful suggestion. In the results-section, we have tried to differentiate between 6b en 6c (child as a whole and child as a person) by rephrasing the titles of the subtopics. We have incorporated your suggestion of the continuum in the discussion section of the manuscript. The added text is: “Medical detachedness versus attention for persons is not black and white, but rather a continuum, ranging from patient-physician relationships that focus on a very specific symptoms, without much attention for the person who is experiencing it, towards recognizing the complexity of the child’s overall health situation, and finally evaluating care in the context of a child who as a living person experiences its illness. In different forms of medicine different levels of detachedness, and attention for the person may be appropriate. A dental appointment, where the physician-patient relationship is short, and the treatment limited to a specific expertise, may call for a different kind of professionalism than paediatric palliative care, where its very goal implies attention for the person.” (see p.17)
Page 19, line 14: The paper ends suddenly after the strengths and limitations. The authors may wish to add a final section on overall implications and conclusions.	Thank you for this feedback. We have added a conclusion-section to the manuscript.
Reviewer: 2 Marjorie S Rosenthal	
ABSTRACT AND INTRO: well written and clear	We thank the reviewer for these kind words.
METHODS: please expand upon your description of	Thank you for these helpful suggestions.

methods. Who wrote the interview guide? What was their expertise? Was it different for different diseases or different respondents? What was the interview guide based on/adapted from?	In addition to the COREQ guideline that was uploaded with the first version of the manuscript, we have improved the methods section to meet both COREQ and SRQR guidelines for reporting qualitative research. Both are added as supplementary files to the manuscript.
Who was on the analysis team? What was their expertise?	The analysis team consisted of a PhD student, an ethicist, a professor in end-of-life care, and a child psychiatrist and a physician/professor in paediatric palliative care. We added this information to the methods section (see p.6) The new text is: The research team (the authors of this paper) consisted of a PhD student, an ethicist, a professor in end-of-life care, and a child psychiatrist and a physician/professor in paediatric palliative care.
Was there an opportunity to share the themes with the parents who were interviewed?	We informed participants by newsletter about the progress of our research. Parents were represented by parents in the advisory board who gave feedback on the research and its findings. We added information on the updating of parents in the methods section. The new text is: "Participants were updated on the progress of the study and findings by e-mail." (See p.6)
How many people were on the advisory board? Did people get paid to participate in it?	There were 9 people on the advisory board. They were offered remuneration for their efforts. We added this to the manuscript. The new text is: "During the study, biannual meetings were held with an advisory board (9 members) of parents, physicians and researchers who provided feedback on the findings. They were offered remuneration for their efforts." (see p.6)
How did the authors settle on the sample size they did? Was there any rationale?	The sample size was determined by data saturation. We continued interviewing until both data saturation for a group with maximum variety was reached. We added this information to the methods section. the added text is: "Recruitment continued until thematic saturation and saturation of variety in participants was achieved." (see p.4)
RESULTS: Overall the results were clear and the headings leading to the quotations were clear. What the authors should do to make this section much more informative would be for them to declare a theme--not just stating that communication, decision-making etc is a barrier but be more specific in the heading--tell the reader your analysis of what the actual issue is. Otherwise, the analysis appears to be superficial--all you did was group the quotations by topic, you didn't try to get to the	We thank the author for this excellent suggestions. We have rewritten the topics into themes. The barriers are now described as follows:  1) parents wished for more empathic and open communication about the illness and prognosis. 2) organizational barriers create bureaucratic obstacles and a lack of continuity of care. 3) parents wished for more involvement in decision-making. 4) parents wished they had more support from the health

themes of the quotations. For example, something like "Parents wished they had more support from the health care team on end-of-life decision making" is more informative than "End-of-life decision-making" The authors should do this for all 6 themes and make Table 4 reflect this as well.	care team on end-of-life decision-making. 5) parents experienced a lack of attention for the family during the illness and after the death of their child. 6) parents experienced an overemphasis on symptom-treatment and lack of attention for their child as a person. These descriptions are used throughout the manuscript to provide more unity.
The authors use quantitative language in the results section for example, "a majority of the parents," When, in fact, that's inappropriate to do in qualitative research--it is mis-leading to the reader bc the sample was not created for that purpose--it wasn't created for quantitative reasons or through quantitative means.	Thank you for bringing this to our attention. Throughout the text we have made modifications and rewritten sentences to avoid quantitative labelling of results. (See results section.)
Is there a Table 3?	Thank you for spotting this mistake. We have renamed table 4 to table 3. Upon request of the other reviewer, the table has been moved to the results section.
TABLE 2The n for maternal education should be n=21. Same for paternal	Thank you for bringing this to our attention. In the interview, we asked for the educational level of both parents. We considered the educational level of both parents, even if one of them was not interviewed, to be possibly of influence on the care and decisions of the child. To avoid confusion, we added a footnote to the table (see table 2)
Page 2, line 7--should be "barriers, as perceived by parent TO good care..."	Thank you for his feedback. We have altered the text to match your feedback. The new text is: “To identify barriers, as perceived by parents, to good care for children with life-threatening conditions”
Page 4, lines 5-7--this sentence is unclear: "Healthcare providers describe as the main barriers is palliative care: communication, an uncertain prognosis, time constraints and lack of education"	We have rewritten the sentence for more clarity. The new text is: The main barriers in paediatric palliative care as experienced by healthcare providers are communication, the uncertainty of the prognosis, time constraints and lack of education. (See p.4)
Page 6, line 43--"Participants were given the opportunity to revise their manuscripts." This sentence is unclear. What do you mean by "manuscript"?	We mistakenly wrote manuscripts instead of transcribed interviews. We have corrected this mistake. The new text is: “Participants were given the opportunity to revise their transcribed interviews.” (See p.5)
Page 11, line 33--What is a "subsidy provider"?	With subsidy providers we wanted to describe the organizations financing care for these children (governments etc.) We have altered the text to ‘financers’ for more clarity. The new text is: “Several parents expressed the need for a case manager, who could bridge the gaps between parents, different healthcare providers and financers” (see p.10)

VERSION 2 – REVIEW

REVIEWER	Douglas Hill The Children's Hospital of Philadelphia, United States of America
REVIEW RETURNED	06-Mar-2020

GENERAL COMMENTS	This is a qualitative study of 64 parents of 44 children with life threatening conditions in the Netherlands. The first author conducted in-person interviews that included parent perceptions of perceived barriers to care. The authors identified 6 barriers to care mentioned by parents in the interviews. First, parents wished for more empathetic and open communication about the illness and prognosis. Second, organizational barriers create bureaucratic obstacles and a lack of continuity of care. Third, parents wished for more involvement in decision-making. Fourth, parents wished they had more support from the health care team on end-of-life decision-making. Fifth, parents experienced a lack of attention for the family during the illness and after the death of their child. Sixth, parents experienced an overemphasis on symptom-treatment and lack of attention for their child as a person. The authors conclude that barriers identified by parents focused almost exclusively on non-medical aspects of care for children with a life-threatening condition suggesting that current care focuses too much on symptoms and not enough on the human beings behind those symptoms. The authors discuss the balance between empathy and professionalism in pediatric palliative care. This is an interesting paper that clearly describes some of the key sources of distress and frustration for parents of children with life threatening illness. While some of the findings are familiar from earlier research, the current study shows these issues are prevalent across the range of life-threatening illnesses, not just cancer or end of life. The authors also report findings of a desire on the part of some parents for more active forms of euthanasia for children this age instead of the current options in the Netherlands of palliative sedation and withdrawal of treatment. In this revision authors have done an excellent job of addressing the issues I raised in my comments and the themes are now presented very clearly in the results. I have just a couple of minor comments on this version that the authors may wish to address at the editor's discretion. Introduction: Page 5, lines 15-17: The authors removed the phrase I thought was unclear: "decisions about the proportionality of life-prolonging treatments and sometimes end-of-life decisions." The revised sentence is an improvement but is still a little confusing: "Care for these children is further complicated because decisions that may limit the life-expectancy of a child (including end-of- life
--

	decisions) are considered part of palliative care [3, 4]” The current wording also seems to support the common belief that palliative care usually results in reduced life expectancy for the child. The authors may wish to rephrase this to clarify that only a subset of palliative care decisions would result in a reduced life expectancy: “Care for these children is further complicated because some decisions to reduce the child’s suffering in the context of palliative may also limit the life-expectancy of a child (such as discontinuing or reducing life-sustaining technology) [3, 4]” Discussion: page 18, line 19-23: the authors refer to “The perceived need for end of life decisions . . .” The authors may want to describe this as “The perceived need for options for active end of life decisions (e.g. euthanasia) . . .” to distinguish this unique finding from the broad category of any end of life decision.
--	---

VERSION 2 – AUTHOR RESPONSE

Reviewer 1 In this revision authors have done an excellent job of addressing the issues I raised in my comments and the themes are now presented very clearly in the results. I have just a couple of minor comments on this version that the authors may wish to address at the editor's discretion.	Thank you very much for these kind words, we are very pleased to hear that our revisions have met your expectations! We also appreciate your additional feedback, and have used it to further improve our manuscript. We will discuss our response in more detail below.
Introduction: Page 5, lines 15-17: The authors removed the phrase I thought was unclear: “decisions about the proportionality of life-prolonging treatments and sometimes end-of-life decisions.” The revised sentence is an improvement but is still a little confusing: “Care for these children is further complicated because decisions that may limit the life-expectancy of a child (including end-of- life decisions) are considered part of palliative care [3, 4]” The current wording also seems to support the common belief that palliative care usually results in reduced life expectancy for the child. The authors may wish to rephrase this to clarify that	Thank you for this feedback. We agree with you that our phrasing suggests that these decisions always lead to a reduced life-expectancy. We think your suggestion is an excellent one, and have used it in our manuscript. The new text is: Care for these children is further complicated because some decisions to reduce the child’s suffering in the context of palliative may also limit the life-expectancy of a child (such as discontinuing or reducing life-sustaining technology). (See p.5)

only a subset of palliative care decisions would result in a reduced life expectancy: “Care for these children is further complicated because some decisions to reduce the child’s suffering in the context of palliative may also limit the life-expectancy of a child (such as discontinuing or reducing life-sustaining technology) [3, 4]”	
Discussion: page 18, line 19-23: the authors refer to “The perceived need for end of life decisions . . .” The authors may want to describe this as “The perceived need for options for active end of life decisions (e.g. euthanasia) . . .” to distinguish this unique finding from the broad category of any end of life decision.	Thank you for this excellent suggestion. We have used your suggestion to improve the text for added clarity, with the addition of “active life-ending on request of the parent, because in the Dutch context, the term euthanasia is only used to refer to active life-ending on request of the patient. The new text is: “The perceived need for options for active end of life decisions (e.g. euthanasia, or active life-ending on request of parents) is a barrier that, to our knowledge, has not previously been mentioned in studies on barriers in care for children with life-threatening conditions.” (see p.17)